# Electrostrictive microelectromechanical fibres and textiles

Tural Khudiyev[1,2], Jefferson Clayton[3], Etgar Levy[1,2], Noémie Chocat[3], Alexander Gumennik[1,2], Alexander M. Stolyarov[4], John Joannopoulos[1,2,5] & Yoel Fink[1,2,3]

Microelectromechanical systems (MEMS) enable many modern-day technologies, including actuators, motion sensors, drug delivery systems, projection displays, etc. Currently, MEMS fabrication techniques are primarily based on silicon micromachining processes, resulting in rigid and low aspect ratio structures. In this study, we report on the discovery of MEMS functionality in fibres, thereby opening a path towards flexible, high-aspect ratio, and textile MEMS. The method used for generating these MEMS fibres leverages a preform-to-fibre thermal drawing process, in which the MEMS architecture and materials are embedded into a preform and drawn into kilometers of microstructured multimaterial fibre devices. The fibre MEMS functionality is enabled by an electrostrictive P(VDF-TrFE-CFE) ferrorelaxor terpolymer layer running the entire length of the fibre. Several modes of operation are investigated, including thickness-mode actuation with over 8% strain at $25\,MV\,m^{-1}$, bending-mode actuation due to asymmetric positioning of the electrostrictive layer, and resonant fibre vibration modes tunable under AC-driving conditions.

[1] Research Laboratory of Electronics (RLE), Massachusetts Institute of Technology, Cambridge, MA 02139, USA. [2] Institute for Soldier Nanotechnologies, Massachusetts Institute of Technology, Cambridge, MA 02139, USA. [3] Department of Materials Science and Engineering, Massachusetts Institute of Technology, Cambridge, MA 02139, USA. [4] Lincoln Laboratory, Massachusetts Institute of Technology, Lexington, MA 02420, USA. [5] Department of Physics, Massachusetts Institute of Technology, Cambridge, MA 02139, USA. Tural Khudiyev and Jefferson Clayton contributed equally to this work. Correspondence and requests for materials should be addressed to Y.F. (email: yoel@mit.edu)

Although the potential for miniature machines was appreciated as early as the 1960s, most famously in Richard Feynman's seminal lecture[1] "There is plenty of room at the bottom", the key enabler of microelectromechanical systems (MEMS) proved to be the vast knowledge on silicon processing technologies[2] developed for the integrated circuit industry in the 1970s and 1980s. By applying silicon micromachining technology to mechanical devices such as cantilevers and membranes, researchers have been able to fabricate increasingly sophisticated miniaturized electromechanical transducers. Today, MEMS have extended the applications of electromechanical transduction well beyond traditional actuation and motion sensing into new fields such as inkjet printing, accelerometers, drug delivery, and projection displays[3]. However, while silicon has been matured for high-throughput MEMS fabrication, the rigidity of conventional Si substrates presents limitations, particularly for non-planar, conformable actuation. Textiles on the other hand are conformable yet to date serve primarily passive functions. In this study, we present a textile MEMS enabled by all-fibre flexible MEMS devices, opening a path towards large-area, conformable, and weavable electromechanical systems.

In the past decade, a new approach for drawing multimaterial polymer-clad fibre devices from preforms has emerged, where fibres are not used as simple longitudinal conduits but instead as transverse devices that operate radially from their surface[4]. The thermal drawing process offers a scalable and controllable means of producing kilometers of uniform functional fibre with inner or outer features of sub-micron dimensions[5–8]. This approach has led to the successful development of optical, optoelectronic, electronic, and thermal fibre devices[4], and presents an opportunity to realize MEMS in a novel form.

Here we report a novel thermally drawn MEMS fibre device based on electrostrictive P(VDF–TrFE–CFE) ferrorelaxor terpolymer. Electromechanical actuation capabilities of this fibre device are established using high voltage atomic force microscopy (HVAFM) and strain values as high as $> 8\%$ are demonstrated. For a fibre with a free length of 3.5 cm and an asymmetric geometry with respect to placement of the electrostrictive layer, a maximum transverse deflection of $\sim 80\,\mu m$ under an applied voltage of 200 V DC is established using contact profilometry. Furthermore, by applying AC fields, frequency and amplitude-tunable cantilever-like resonant vibrations are observed in the fibre. We use this AC-driving scheme to demonstrate a fibre optical modulator, which is used to deflect an optical beam incident transverse to the fibre axis. Modulation of incident light through electric field induced deflection is demonstrated up to the second harmonic of the fibre at 158.3 Hz for a fibre that is $\sim 6$ cm long. Moreover, we demonstrate a hybrid fibre device woven into a textile consisting of a surface sub-wavelength photonic structure and an internal MEMS domain.

## Results

**Fabrication of electrostrictive fibre MEMS.** The primary design consideration for producing a high-performance MEMS fibre device concerns the selection of an electromechanical

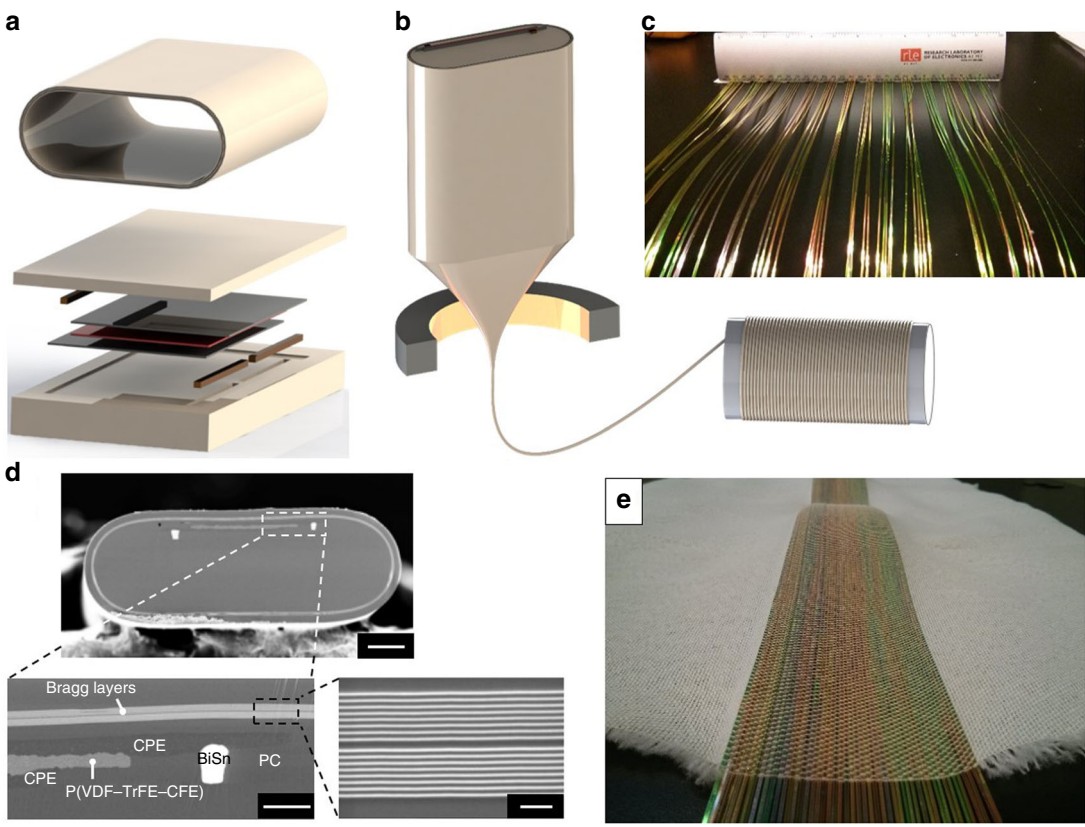

**Fig. 1** Fabrication of electrostrictive fibres. **a** Schematic of the preform assembly for a multimaterial electrostrictive fibre. A P(VDF-TrFE-CFE) layer (red) is assembled with CPE polymer electrodes (black), Bi–Sn electrodes (brown) and a PC cladding (beige), and consolidated. The surrounding shell incorporates a multilayer $As_{25}S_{75}$/PC structure (top). **b** Schematic of the preform-to-fibre draw process. **c** Array of flexible electrostrictive fibres shows colored reflections via the Bragg effect. **d** SEM micrographs of the overall structure and close-up of a multimaterial electrostrictive fibre and Bragg layers. Scale bars for top, bottom left, and bottom right are 100, 20, and 2 μm, respectively. **e** We demonstrate the capability to integrate our fibre MEMS into the textile using conventional weaving machines

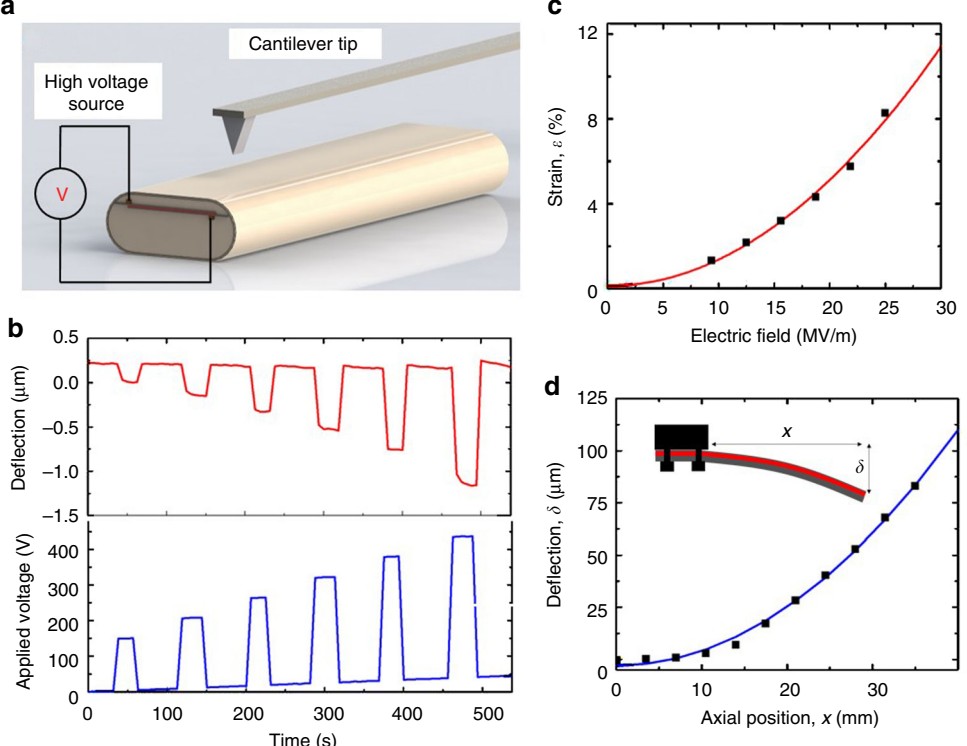

**Fig. 2** Characterization of electrostrictive fibres. **a** Experimental set-up for HVAFM and contact profilometer measurements. **b** AFM cantilever tip displacement at the surface of the fibre under increasing voltage. **c** Electrostrictive strain in the fibre calculated from tip displacement and applied electric field in the terpolymer layer. The solid red line is a second order fit for the measured data and well-agreed with the electrostriction principle (i.e., quadratic dependence of strain to the applied electric field). **d** Contact profilometry measurement of fixed-end fibre deflection at 200 V. The solid blue line is a quadratic fit for the measured data and agreed with the beam deflection case (i.e., quadratic dependence of deflection to the fibre length) where cantilever (fibre in our case) is subjected to bending moment

transduction mechanism and materials compatible with thermal fibre drawing. Piezoelectric and electrostictive actuation mechanisms are two potential candidates for achieving electro-mechanical transduction in polymer-clad fibres. While electro-striction was long considered a higher-order nonlinear effect requiring stronger electric fields than typically used for piezo-electrics, the emergence of a new class of highly electrostrictive materials, known as relaxor ferroelectrics, has opened the path to the use of electrostriction as an alternative to the piezoelectricity for the large-strain actuation applications[9]. In addition to several percent maximal achievable strain capabilities, a major advantage of the electrostriction over the ferroelectric mechanism is that it enables a reproducible, non-hysteretic response. Furthermore, unlike ferroelectric piezoelectrics, electrostrictors do not require high-field electrical poling, and are more stable over time. Owing to these properties, in this work we take the advantage of the electrostriction effect as an efficient electromechanical transduc-tion mechanism and employ a relaxor ferroelectric material, P (VDF–TrFE–CFE), known for exhibiting one of the largest elec-tromechanical strain among PVDF-based terpolymers[10]. The melting temperature of P(VDF–TrFE–CFE) is 125 °C. The other materials comprising the device architecture are selected such that all materials can flow at a common temperature when thermally drawn.

The fabrication of multimaterial fibre devices starts with constructing a preform, which is a macroscopic scaled-up version of the fibre, with the same composition and cross-sectional structure as the final device. This preform is then heated in a cylindrical furnace and drawn into hundreds of meters of fibre. The key to this process is the identification of a set of materials that can be co-drawn, while conserving the device architecture

from preform to fibre by preventing capillary break-up and mixing due to flow instabilities. Viscous forces are commonly employed to kinetically avert these surface–tension driven phenomena[9]. In particular, the use of the viscous conducting materials such as carbon black polymer composites has been reported for the integration of large aspect-ratio electrodes in fibres[11]. Previous work has demonstrated that ferroelectric P (VDF–TrFE) can be thermally drawn in a polycarbonate (PC) cladding with carbon-loaded polyethylene (CPE) electrodes[6]. Given that P(VDF–TrFE–CFE) has a melting temperature that is 25 °C lower than that of P(VDF–TrFE), it can be easily co-drawn with the same materials system. We fabricate a preform by contacting a 300 μm-thick layer of P(VDF–TrFE–CFE) with CPE electrodes in a parallel-plate capacitor configuration and embed this assembly within a polycarbonate (PC) matrix (Fig. 1a, see Methods section for details). The small metallic buses of the low melting temperature Bi–Sn alloy ($T_m = 138$ °C) are inserted adjacent to the CPE to facilitate long-range electrical transport along the length of the fibre. A multilayer structure composed of PC and $As_{25}S_{75}$ glass are wrapped on the outer surface of the preform. When drawn, these layers reduce in thickness leading to a 1D photonic bandgap structure providing surface reflection from the extended surface area of the fibre to facilitate the deflection of the optical beams incident transverse to the fibre axis. The preform is then consolidated in a vacuum oven and drawn into extended lengths of the fibre using the draw procedure described in the Methods section (Fig. 1b, c). SEM micrographs of the cross-section of a multimaterial electrostric-tive fibre show that the device structure is well preserved during the draw (Fig. 1d). The materials show good adhesion and the sharp angles in the device architecture are maintained. We have

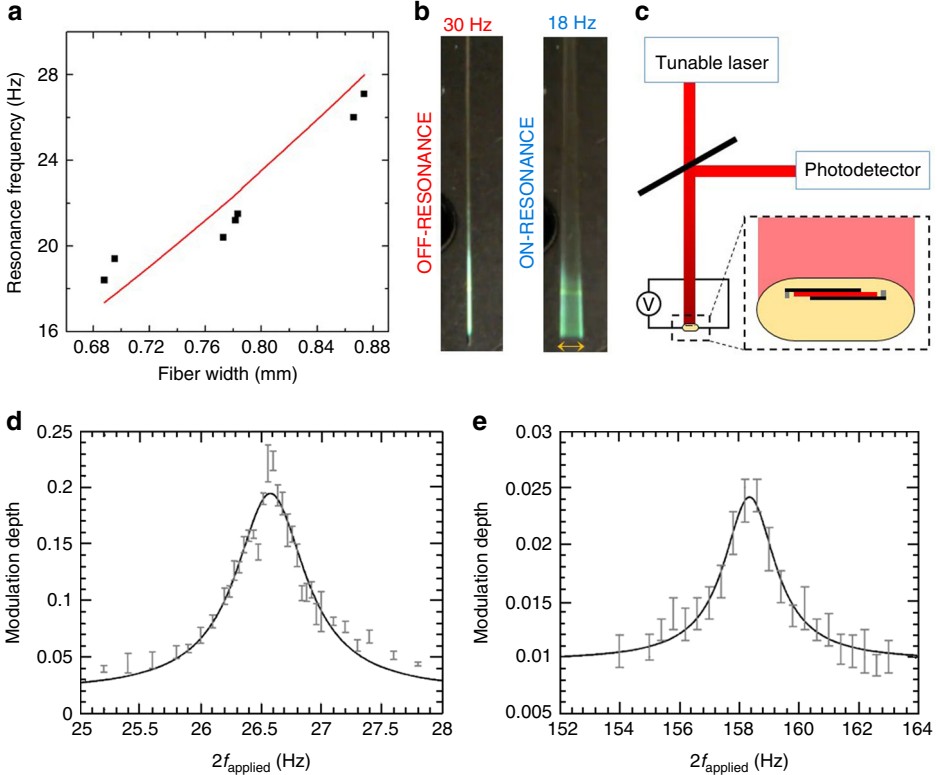

**Fig. 3** Electrostrictive fibre resonances. **a** Fiber width is utilized to adjust resonance frequency of the fibre MEMS. **b** Amplitude of oscillation is shown both for on and off-resonance frequency points. **c** Optical setup to measure amplitude modulation under applied voltage and driving frequency. **d** Modulation depth measured in the vicinity of the first harmonic, approximately 26.6 Hz. Solid line is a fitted Lorentzian curve. **e** Modulation depth measured in the vicinity of the second harmonic, ~158.3 Hz. Solid line is a fitted Lorentzian curve. Error bars in **d**, **e** represent a width of one standard deviation about the mean of the measured sample set, which contains three measurements for each frequency point

demonstrated that the drawn fibres can be treated as textile fibres and can be woven using the same equipment used to process conventional textile fibres (Fig. 1e, Supplementary Fig. 1). MEMS behavior of the fibres is confirmed after the weaving process. Embedding MEMS actuation in a textile form factor can enable active fabric materials which are conformable, soft and can cover large areas.

**Characterization of fibre MEMS in thickness-mode.** Next we characterize the performance of the fibre MEMS in thickness-mode. When a voltage is applied across the CPE electrodes, the P (VDF–TrFE–CFE) layer contracts in the thickness direction and expands in the lateral direction. The amount of electrostrictive strain achievable in these fibres is measured using a technique based on high voltage atomic force microscopy (HVAFM). Compared with traditional techniques where the voltage is applied via the AFM tip, HVAFM has the advantage of enabling higher and more uniform applied electric fields, and its use has been demonstrated for piezoelectric coefficient measurements[12]. The experimental setup is shown in Fig. 2a. The fibre sample is affixed in epoxy on one face to a glass slide, connected to an external DC high voltage power source, and placed under an AFM tip in a contact mode. Figure 2b shows the AFM tip displacement at the surface of the electrostrictive fibre as increasing voltage steps are applied to the fibre through the external power source. The slight data spread at high fields is due to the relaxation phenomena in the fibre, which results in different step heights when the voltage is turned on and off. The observed tip displacement is consistent with electric field-induced contraction of the electrostrictive layer. This measurement is repeated on several samples with identical dimensions where similar

displacements are observed for all fibres at each voltage step. The measured step heights are plotted against the applied voltage, both normalized by the average P(VDF–TrFE–CFE) layer thickness (Fig. 2c).

Electrostriction phenomenon is considered a higher-order nonlinear effect where strain exhibits a quadratic dependence on the applied electric field;

$$\varepsilon = ME^2 \qquad (1)$$

where $\varepsilon$ represents strain, $E$ is applied electric field and $M$ electrostriction coefficient. Our experimental findings clearly satisfy the quadratic relation between strain and electric field for fields up to 25 MV m$^{-1}$. A maximum strain of > 8%, calculated from known terpolymer layer thickness and tip displacement, is achieved for an applied electric field of 25 MV m$^{-1}$; this corresponds to a contraction of 1.3 μm for a 16 μm-thick terpolymer layer. While > 7% percent of lateral strain has been reported under ~ 150 MV m$^{-1}$ for free-standing P(VDF–TrFE–CFE) films[13], the electrostrictive strain in fibre form exhibits remarkably better performance under significantly lower electric fields. Note that this strain value is also around two orders of magnitude larger than strains achievable with piezo-electric polymers, which are on the order of 0.1%. We derive the electrostriction coefficient in the thickness direction ($M_{33} = 1.28 \times 10^{-16}$ m$^2$ V$^{-2}$) by performing a second order polynomial fit of measured data. In general, electrostriction coefficients of relaxor ferroelectric materials are highly dependent on the particular processing conditions[13–16] such as heat treatment, stretching, structure dimensions, or external irradia-tion. In our case, the electrostriction coefficient is one order of

magnitude higher relative to previously reported value for PVDF–TrFE–CFE film prepared via solution-based processing. We attribute this difference to lower material thickness and both heat-treatment and stretching processes which are present during the fibre drawing process. Higher coefficient provides the ability to achieve much higher strains for a given field. We expect even higher strain rates to be potentially achievable by changing the fibre design such that the P(VDF–TrFE–CFE) layer would be constrained in the lateral direction by a softer material, for example low density polyethylene or a thermoplastic elastomer which can have a Young's modulus at least one order of magnitude smaller than polycarbonate. The driving voltage required to achieve a given strain can also be further lowered by adopting a stacked-layer geometry[6].

**Characterization of fibre MEMS in bending-mode.** The preform design space permits the placement of the electromechanical transducer anywhere within its cross-section. By deliberately introducing this layer off-centre, asymmetric strain fields can be induced in the fibre, leading to the emergence of transverse-deflection bending modes. We investigate this phenomenon using contact profilometry. The experimental setup is shown in Fig. 2a, bearing resemblance to the HVAFM measurement setup but with cantilever tips designed for larger displacements. For an applied voltage of 200 V DC and stylus pressure of 2 mg, transverse deflection (δ) is measured along the axial direction ($x$) of a 3.5 cm fibre fixed flat on one end to a silicon wafer. The measurement is repeated on several fibres of the same length and the results are shown in Fig. 2d. The deflection profile of the fibre is well-fit into a quadratic shape as expected from the following beam deflection case:

$$\delta = \frac{Mx^2}{2EI} \qquad (2)$$

where $M$ is the bending moment, $E$ is Young's modulus, and $I$ is the moment of inertia of the cross-section about its neutral axis. This quadratic profile is originated from the balancing of the bending moment generated by the misfit strain (i.e., due to electrostrictive strain) against the opposing moment offered by the fibre cladding. A maximum transverse displacement of ~ 80 μm is measured for a fibre fixed on one end and with a free length of 3.5 cm; the fibre is observed to bend opposite to the face containing the electrostrictive device (see Supplementary Movie 1). We use beam deflection equations to derive the electrostriction coefficient $M_{31}$ for PVDF-TrFE-CFE material, which is found to be around ~ $10^{-17}$ m$^2 \cdot$ V$^{-2}$ (Supplementary Fig. 2). This asymmetry in the placement of the electrostrictive device is shown to be key in inducing a bending behavior that resembles a cantilever.

**Resonant mode of fibre MEMS.** The above-described modes of operation both utilized DC driving fields. By driving the fibre under AC voltage, the electromechanical energy conversion process becomes frequency-dependent and can take advantage of resonance effects. Unlike purely longitudinal vibrations which would be expected from a free-standing electrostrictive layer, here by sandwiching the electrostrictive material between cladding material in a fibre, we shift the direction of vibrations and produce cantilever-like flexible MEMS device that can oscillate at resonance frequencies of the fibre. To this end, the one edge of the fibre is fixed while another is allowed to vibrate. The resonance characteristics, which are based on the fibre dimensions, are modeled using beam harmonics equations (see Methods section). Since the fibre-drawing technique allows us to alter fibre dimensions during the fabrication stage, we can characterize a

range of fibre sizes and produce flexible fibre MEMS resonators over a broad spectrum of frequencies. We explore the resonance effect with several fibres of fixed lengths and distinct cross-sectional dimensions and show how the resonance frequency increases with increasing fibre width (Fig. 3a, see also Supplementary Movie 2). In particular, the amplitude of the oscillations at 18 Hz (on-resonance) and 30 Hz (off-resonance) frequency points can be clearly observed from Fig. 3b (see also Supplementary Movie 3). It is also possible to control the magnitude of the oscillation from nm-scale to cm-scale by altering the magnitude of the applied electric field. We demonstrate the tuning of the resonance amplitude for a 7 cm-long fibre by changing the voltage between 0 and 350 V for a fixed resonance frequency of 20 Hz (see Supplementary Movie 4).

Having established that the strain in the electrostrictive layer results in transverse deflection of the fibre, we proceed to illustrate and characterize the harmonic properties of an integrated photonic MEMS fibre system—a fibre that contains both a Bragg mirror and an electrostrictive device. This system is demonstrated in the context of optical beam deflection. Here the fibre plays a dual role—the Bragg structure reflects an incoming optical beam, while the MEMS structure controls the deflection angle. To measure the deflection of the incident optical beam, a laser is directed onto the face of a fibre that contains a photonic mirror (i.e., a Bragg layers designed to reflect at the wavelengths of 1300–1800 nm). The signal reflected back from the fibre surface is detected by a photodetector and recorded with an oscilloscope. The fibre is driven by an external high voltage power amplifier and function generator (Fig. 3c, see Methods section for details). The measurement is performed by maintaining a fixed applied voltage amplitude and sweeping the voltage driving frequency while recording the photocurrent at the detector. The optical beam deflection (and associated optical flux incident at the detector) is registered as a modulation depth (($V_{max}-V_{min}$)/$V_{avg}$) in the photocurrent at the detector. The greatest modulation depth coincides with the resonance frequency point, where the fibre vibration amplitude is the largest. We measure a maximum modulation depth of 22.5% at the first harmonic (Fig. 3d) with an applied voltage of 90 V, and 2.5% at the second harmonic (Fig. 3e) with an applied voltage of 150 V. Note that although the magnitude of the modulation depth is affected by the distance between the fibre and the detector, the Lorentzian line shape which characterizes the harmonic behavior of the fibre is invariant to the experimental setup. The experimentally extracted first and second harmonic frequencies were 26.58 ± 0.02($2\sigma$) (Hz) and 158.3 ± 0.1($2\sigma$) (Hz), respectively, and they are shown to fall within the predicted values of 24.3 ± 2 (Hz) and 152 ± 10 (Hz), respectively. The harmonic frequencies were predicted using solutions from the well-known Euler–Bernoulli beam theory (see Methods section for details), and the range of values in the predicted harmonic frequencies is dominated by the uncertainty in the Young's modulus of the composite fibres.

The above-described work focused on individual fibre characterization and highlights the novelty in the degrees of freedom available in the fibre design. Extending the individual fibre case to a woven structure or fibre array further extends the capabilities to the level of an integrated fibre system, paving the way towards more complex functionality. We illustrate this potential by embedding fibres in a polydimethylsiloxane (PDMS) matrix and using the resulting composite (Supplementary Fig. 3a) to deflect an incident optical beam at low frequencies. Modulation depth is measured using the previously described single fibre setup, and incident light is focused on the centre of the fibre array. The output of the photodetector over time for a driving frequency of 1 Hz and an applied voltage of 300 V is shown in the lower half of Supplementary Fig. 3b. The top half is an illustration

of the relationship between driving frequency and fibre response frequency for an electrostrictive material, showing the characteristic frequency-doubling effect that can distinguish electrostriction from other forms of electromechanical transduction, such as piezoelectricity[9].

Besides the surface modulation, we also show the potential for direct modulation of the sub-wavelength photonic cavity in the fibre. Application of an electric field results in spectral tuning from the direct compression (or expansion) of the multilayer structure (Supplementary Fig. 4). The wavelength shift is mainly influenced by the stiffness of the cladding material. Larger spectral shifts can be achieved by employing more elastic cladding materials such as elastomers and can be useful for the applications such as dynamical color tuning or spectral filtering.

## Discussion

We note several potential future research directions and potential applications based on the unique fibre MEMS features. First, the axially symmetric and indefinitely long fibre platform can open the door for MEMS textiles, initiating novel opportunities in advanced functional fabrics, such as holographic display technologies[17]. Furthermore, electrically controllable microfluidic pumps[18] can be enabled by integrating electrostrictive polymer devices with microfluidic channels in fibres[19], which present novel on-demand material (e.g., drug, solvent) release schemes in textiles.[20] Second, the realization of long, thin, and flexible fibres with electromechanical transduction capabilities could enable new sensing and actuation applications in inaccessible regions or over extended lengths. For instance, the demonstrated actuation mechanism can provide the ability to navigate through the branches of narrow lumens of the body. Furthermore, other functions (such as sensing) can be integrated into the same fibre while keeping the diameter of the device as low as required paves the way for the development of novel types of steerable catheters. Third, the ability to assemble fibre devices into grids or arrays makes them particularly well-suited for the coverage of large non-planar surfaces[11,21], an important feature for applications such as solar energy harvesting systems[22,23]. For instance, flexible fibre MEMS with low power consumption can be embedded into flexible polymeric solar panels with integrated solar tracking capabilities. Finally, monolithic integration of electrodes into the fibre enables straightforward connection with external electrical circuits[24]. This could enable electrically controllable artificial muscles which resemble human muscles, especially if elastomeric materials are utilized in the cladding of fibre MEMS.

We have demonstrated the first integration of a relaxor ferroelectric polymer in a thermally drawn fibre for electromechanical actuation. Strain levels of >8% are measured for P (VDF–TrFE–CFE) material in the fibre device using high voltage atomic force microscopy and a maximum transverse deflection of 80 μm is demonstrated for 3.5 cm-long fibre under an applied DC voltage of 200 V via contact profilometry. The potential of this approach to realize complex electromechanical systems in fibres is illustrated by the fabrication of a hybrid photonic electrostrictive device capable of deflecting a laser beam reflecting off the surface of the fibre. Deflection of incident light through electric field induced fibre actuation is demonstrated up to the second harmonic frequency of the fibre at 158.3 Hz.

## Methods

**Electrostrictive fibre fabrication**. The P(VDF–TrFE–CFE) terpolymer is purchased in powder form from Piezotech S.A.S. (France) and melt-pressed at 155 °C under 110 bars to form 300 μm-thick films. It is then assembled into a preform with 300 μm-thick CPE films, eutectic $Bi_{58}Sn_{42}$ electrodes ($T_M = 138$ °C, Indium Corporation), and PC bar cladding (McMaster) using traditional milling techniques. The preform is consolidated in a vacuum oven at 185 °C for 20 min to remove trapped gas and form high quality interfaces. This final preform is 38 mm wide, 11 mm thick and 200 mm long. It is then thermally drawn in a three-zone vertical tube furnace with the top-zone temperature at 150 °C, the middle-zone temperature at 230 °C, and the bottom-zone temperature at 110 °C. The preform is fed into the furnace at a speed of 1 mm min$^{-1}$, and the fibre was drawn at a speed of 0.8–2.0 m s$^{-1}$. The tension during the draw is 100–200 g, corresponding to an average stress of 2–4 kPa. With this procedure, meters of fibres are drawn with thicknesses ranging from 240 to 460 μm (widths ranged from 680 μm to 1.3 mm).

**High voltage atomic force microscopy**. HVAFM measurements is performed with a Dimension 3100 Scanning Probe Microscope in contact mode with Nanoscope V control station (Veeco). The AFM tip contacts on the fibre surface at a frequency of 1 Hz, with the slow-axis scan disabled. The fibre is connected to an external DC power source (Stanford Research PS350). There is an additional connection between DC power source and AFM to record synchronized data of voltage and deflection.

**Contact profilometry**. Deflection measurements are performed with a Sloan Technology Corporation Dektak III surface profilometer using a 2 mg stylus pressure. The fibre is fixed in epoxy on one end to a silicon wafer and the profilometer scanned a 5 mm line in a direction perpendicular to the axis of the fibre (i.e., across its width). The fibre is connected to an external DC power source (Stanford Research PS350).

**Optical measurement**. A laser (ANDO AQ4321D—tunable laser with wavelength range of 1520–1620 nm) is directed on the Bragg surface of the fibre. The signal reflected back from the fibre is detected by a photodetector (Thorlabs PDA10CS amplified InGaAs photodetector 700–1800 nm) and recorded with an oscilloscope connected to a computer. The fibre and photodetector are placed at a distance $d = 11$ cm apart, so that the photodetector is in the image plane of the fibre; for the fibre array in PDMS, the distance $d = 40$ cm. For DC measurements, the fibre is connected to a DC high voltage power supply (Stanford Research PS350). For AC measurements, we use a high voltage amplifier (TREK 10/10B) connected to a function generator. Modulation depth is calculated as signal amplitude divided by mean voltage, and then reported as a percentage.

**fibre Array in polydimethylsiloxane**. Polydimethylsiloxane (PDMS) is purchased from Dow Corning as two-part Sylgard 184 Silicone Elastomer Kit. Base and curing agent are mixed in a ratio of 10:1 and placed under vacuum for 20 min to remove air bubbles. The mix is then poured over fibres that had been overlaid to form an array, and then cured in an over at 80 °C for 1.5 h.

**Analytical derivation of the beam harmonic frequencies**. The analytical derivation is based on the Euler-Bernoulli beam theory. Applying this theory to our fibre, the n-th harmonic frequency of the fibre is given by $f_n = [x_n/(2\pi L^2)]\sqrt{EI/\lambda}$, where $L = 5.8 \times 10^{-2}$ (m) is the fibre free length, $E = 2.3 \pm 0.3$ GPa is Polycarbonate Young's modulus, $I = 4.2 \times 10^{-15}$ (m$^4$) is the fibre second moment of area, $\lambda = 4.6 \times 10^{-4}$ (kg m$^{-1}$) is the fibre mass per unit length, and $x_1 \approx 3.51$ and $x_2 \approx 22.03$.

**Data availability**. The data that support the findings of this study are available from the corresponding author upon reasonable request.

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

## Acknowledgements

This work was supported in part by the MIT MRSEC through the MRSEC Program of the National Science Foundation under award number DMR-1419807 and also was supported in part by the US Army Research Laboratory and the US Army Research Office through the Institute for Soldier Nanotechnologies, under contract number W911NF-13-D-0001. This material is based upon work supported by the Assistant Secretary of Defense for Research and Engineering under Air Force Contract No. FA8721-05-C-0002 and/or FA8702-15-D-0001.

## Author contributions

T.K., J.C., N.C. and Y.F. designed the study. J.C. drew the multimaterial fibres, T.K. performed the AFM and profilometry experiments, and conducted the characterization of the electrostrictive fibres. J.C., E.L., A.G. and A.M.S. have designed the optical setup. J.C. and E.L. made optical measurements. T.K., J.C., A.M.S. and N.C. have written the manuscript. T.K., J.C., E.L., N.C., A.G., A.M.S., J.J. and Y.F. have discussed the results and revised the manuscript.

## Additional information

**Competing interests:** The authors declare no competing financial interests.

