## [Peer Review File · Nature Communications]

Reviewers' comments:

Reviewer #1 (Remarks to the Author):

The authors have presented a very effective way of producing scalable, flexible, high aspect ratio micro-electromechanical systems. In principle, this is a nice idea, though the obvious longitudinal invariance poses a question as to how useful the structure will be. This leads me to my main criticism regarding the paper. The demonstrated application is quite unconvincing.

The experiment is not explained very clearly, however, it seems that the beam is being modulated via a spatial filtering that is associated with a displacement in the fibre. If this is the case this is a rather dull application of what could be a promising technology. For example, I could do this much more effectively with a galvo mirror, which would be wavelength insensitive and would not rely on harmonic oscillation frequencies.

Further to this, the figures given for the modulation depth are near arbitrary. Could I not increase this by moving the detector further from the source?

I would have liked to have seen the Bragg mirror characterised as a function of voltage, if the Bragg wavelength can be modified, this could have led to a much more convincing amplitude modulation demonstration.

The paper certainly has merit and I think would benefit from more complete characterisation of the materials. As a key example, the authors claim a 1 order of magnitude increase in the electrostriction coefficient when compared to previous devices made from the same material. I think that this is significant and should be investigated further. A mere attribution to untested parameters is not scientifically robust enough.

In my opinion the paper needs major revisions with the 'modulation experiment' removed and a more convincing explanation as to the importance of this work before it can be considered for publication in Nature Communications.

Attention to detail is really quite poor. For example.

- 2 spelling mistakes in the abstract
- referencing of figures in wrong order
- spaces between quantity and unit inconsistent
- 3 different types of apostrophe used in Young's
- different degree symbols used

Reviewer #2 (Remarks to the Author):

This is fascinating stuff and I enjoyed reading the paper, as I am sure would many others. I have found no errors (beyond the odd typo) and the work seems to me original and sound. I am also impressed with the various extensions and elaborations you have done.

My remaining issue is really with what you consider your paper to be about. You have clearly demonstrated electrostriction as a practical and interesting phenomenon for MEMs and similar small structures, and I am personally supportive of 'fibre' drawing as a low-cost volume method for making small accurate 'transverse' structures. The latter idea is not novel but you have done a fine implementation.

So this is fine - but if you were seeking to demonstrate that this is a potential engineering

approach to making any of the devices suggested by your application list at the end a lot more is needed. It is not clear that your suggested approach either is or can easily be seen to have the potential to be, the best solution to any of the problems suggested. Even the low-cost-transverse-fab needs attention to packaging for example and the speeds are not high even by the standards of devices this size.

So can you come up with a potential application for which your approach could be a game-changer, and preferably support from some group involved? Fabrics? (for buildings and clothes) for example.

Your paper is still interesting as it stands but could be much stronger with an outline of why.

Reviewer #3 (Remarks to the Author):

This paper is well written, and the results are interesting and convincing. It is a continuation of long standing work by this team, in which quite materials of very different material properties are co-drawn in fibre form. It extends this approach to relaxor ferroelectric materials, and investigates the use of this technology for micro mechanical devices.

While the results are interesting- and potentially may have multiple applications, they are not "new physics". Given the past history of the group (which has been working on co-drawing very different materials since 2000), it is debatable that it is sufficiently ground-breaking/surprising to be published in Nature. It might be more appropriate to a more specialist journal (perhaps Nature Materials).

A technical question: the authors note: "the electrostrictive strain in fiber form exhibits remarkably better performance under significantly lower electric fields" than has been observed previously. They attribute this to processing conditions during the draw, which is a reasonable assumption. This being so, and given the relatively low temperatures used in the draw, it would be interesting to comment on the stability of the effects over time/elevated temperatures/other conditions that may arise in an application.

*Reviewers' comments are in **bold**, the authors' responses are Roman.*

Reviewer #1:

The authors have presented a very effective way of producing scalable, flexible, high aspect ratio micro-electromechanical systems. In principle, this is a nice idea, though the obvious longitudinal invariance poses a question as to how useful the structure will be. This leads me to my main criticism regarding the paper. The demonstrated application is quite unconvincing. The experiment is not explained very clearly, however, it seems that the beam is being modulated via a spatial filtering that is associated with a displacement in the fibre. If this is the case this is a rather dull application of what could be a promising technology. For example, I could do this much more effectively with a galvo mirror, which would be wavelength insensitive and would not rely on harmonic oscillation frequencies.

We thank the reviewer for the thoughtful comments. We agree that in the original manuscript the use of the fibers in a system context was not well articulated. To address this deficiency we have proceeded to add data regarding the incorporation of these fibers into a woven fabric using industrial weaving equipment – demonstrating the first fabric made of MEMS fibers. MEMS behavior is confirmed after the weaving process. In addition, we have modified the introduction and abstract to highlight this new dimension. The purpose of the experiment was not to demonstrate an application but to highlight the novel dimensions of this technology and illustrate the degrees of freedom in the fiber and fabric system design. In particular the integration of a high quality optical Bragg mirror and a MEMS device within a textile fiber sets the stage for how one could realize system level performance within a fiber. This is an enabling capability that could have multiple potential applications in advanced functional textiles, including holographic display technologies^{1,2}, on-demand drug release³ for health and security purposes, etc.

In addition, we thank the reviewer for pointing out that the experimental details regarding the fiber modulation were not sufficient and we have added the following details in the body of the manuscript to address this shortcoming.

¹ Takaki, Y., Holographic 3D display using MEMS spatial light modulator (2012).

² Pearson, E.L., Mems spatial light modulator for holographic displays (2001).

³ Liu, C., Recent developments in polymer MEMS (2007).

“Having established that the strain in the electrostrictive layer results in transverse deflection of the fiber, we proceed to illustrate and characterize the harmonic properties of an integrated photonic MEMS fiber system – a fiber that contains both a Bragg mirror and an electrostrictive device. This system is demonstrated in the context of optical beam deflection. Here the fiber plays a dual role – the Bragg structure reflects an incoming optical beam, while the MEMS structure controls the deflection angle. To measure the deflection of the incident optical beam, a laser is directed onto the face of a fiber that contains a photonic mirror (i.e., a Bragg layers designed to reflect at the wavelengths of 1300-1800 nm). The signal reflected back from the fiber surface is detected by a photodetector and recorded with an oscilloscope. The fiber is driven by an external high voltage power amplifier and function generator (Figure 3c, see Experimental section for details). The measurement is performed by maintaining a fixed applied voltage amplitude and sweeping the voltage driving frequency while recording the photocurrent at the detector. The optical beam deflection (and associated optical flux incident at the detector) is registered as a modulation depth $((V_{max} - V_{min})/V_{avg})$ in the photocurrent at the detector. The greatest modulation depth coincides with the resonance frequency point, where the fiber vibration amplitude is the largest.. ...

... The above-described work focused on individual fiber characterization and highlights the novelty in the degrees of freedom available in the fiber design. Extending the individual fiber case to a woven structure or fiber array further extends the capabilities to the level of an integrated fiber system, paving the way towards more complex functionality.”

2. Further to this, the figures given for the modulation depth are near arbitrary. Could I not increase this by moving the detector further from the source?

Response:

We thank the reviewer for pointing out the deficiency in our explanation. We have made modifications to the manuscript to more clearly explain the figure and the experiment.

“Note that although the magnitude of the modulation depth is affected by the distance between fiber and detector, the Lorentzian line shape which characterizes the harmonic behavior of the fiber is invariant to the experimental setup.”

3. I would have liked to have seen the Bragg mirror characterized as a function of voltage, if the Bragg wavelength can be modified, this could have led to a much more convincing amplitude modulation demonstration.

Response:

We thank the reviewer for this suggestion. In the revised manuscript, we provide experimental evidence in the supplementary section that shows how the applied voltage can change the Bragg cavity spectral bands. To this end, we utilized thickness mode actuator driven by a DC field. This feature can be even more pronounced for applications such as dynamical color tuning or spectral filtering if one of Bragg layer would consist of an elastomer material.

“Besides surface modulation, we also show the potential for direct modulation of the sub-wavelength photonic cavity in the fiber. Application of an electric field results in spectral tuning from the direct compression (or expansion) of the multilayer structure (Figure S4). The wavelength shift is mainly influenced by the stiffness of the cladding material. Larger spectral shifts can be achieved by employing more elastic cladding materials such as elastomers and can be useful for the applications such as dynamical color tuning or spectral filtering.”

4. The paper certainly has merit and I think would benefit from more complete characterization of the materials. As a key example, the authors claim a 1 order of magnitude increase in the electrostriction coefficient when compared to previous devices made from the same material. I think that this is significant and should be investigated further. A mere attribution to untested parameters is not scientifically robust enough.

Response:

We agree that additional investigations on various form factors of the electrostrictive material would be interesting, but we feel it is outside the scope of this work. The electrostrictive coefficient for the material in the undrawn film state is established in the literature and referenced in the manuscript. We demonstrate that by drawing the fiber, we are able to increase the published coefficient value by one order of magnitude and we provide a hypothesis for why this is observed, giving references to support our rationale. Note that we use HVAFM to characterize the electrostrictive coefficients (*i.e.* strain-electric field behavior), which is the most widely used and accurate method for performing this measurement¹.

In my opinion the paper needs major revisions with the 'modulation experiment' removed and a more convincing explanation as to the importance of this work before it can be considered for publication in Nature Communications.

Response:

¹ Li, F., et. al., Electrostrictive effect in ferroelectrics: An alternative approach to improve piezoelectricity (2014).

We believe that the changes described above show the novelty of this work and its relevance to a broad readership. We have also added the following discussion to the conclusion of the manuscript.

“We note several potential future research directions and potential applications based on the unique MEMS fiber features. First, the axially-symmetric and indefinitely long fiber platform can open the door for MEMS textiles, initiating novel opportunities in advanced functional fabrics, such as holographic display technologies. Furthermore, electrically controllable microfluidic pumps can be enabled by integrating electrostrictive polymer devices with microfluidic channels in fibers, which present novel on-demand material (e.g. drug, solvent) release schemes in textiles. Secondly, the realization of long, thin, and flexible fibers with electromechanical transduction capabilities could enable new sensing and actuation applications in inaccessible regions or over extended lengths. For instance, the demonstrated actuation mechanism can provide the ability to navigate through the branches of narrow lumens of the body. Furthermore, other functions (such as sensing) can be integrated into the same fiber while keeping the diameter of the device as low as required paves the way for the development of novel types of steerable catheters. Thirdly, the ability to assemble fiber devices into grids or arrays makes them particularly well-suited for the coverage of large non-planar surfaces, an important feature for applications such as solar energy harvesting systems. For instance, flexible fiber MEMS with low power consumption can be embedded into flexible polymeric solar panels with integrated solar tracking capabilities. Finally, monolithic integration of electrodes into the fiber enables straightforward connection with external electrical circuits. This could enable electrically controllable artificial muscles which resemble human muscles, especially if elastomeric materials are utilized in the cladding of fiber MEMS.”

Attention to detail is really quite poor. For example.

- 2 spelling mistakes in the abstract
- referencing of figures in wrong order
- spaces between quantity and unit inconsistent
- 3 different types of apostrophe used in Young's
- different degree symbols used

Response:

We thank reviewer for these corrections. We corrected them in the manuscript.

Reviewer #2:

This is fascinating stuff and I enjoyed reading the paper, as I am sure would many others. I have found no errors (beyond the odd typo) and the work seems to me original and sound. I am also impressed with the various extensions and elaborations you have done. My remaining issue is really with what you consider your paper to be about. You have clearly demonstrated electrostriction as a practical and interesting phenomenon for MEMs and similar small structures, and I am personally supportive of 'fibre' drawing as a low-cost volume method for making small accurate 'transverse' structures. The latter idea is not novel but you have done a fine implementation.

1. So this is fine - but if you were seeking to demonstrate that this is a potential engineering approach to making any of the devices suggested by your application list at the end a lot more is needed. It is not clear that your suggested approach either is or can easily be seen to have the potential to be, the best solution to any of the problems suggested. Even the low-cost-transverse-fab needs attention to packaging for example and the speeds are not high even by the standards of devices this size. So can you come up with a potential application for which your approach could be a game-changer, and preferably support from some group involved? Fabrics? (for buildings and clothes) for example. Your paper is still interesting as it stands but could be much stronger with an outline of why.

Response:

We thank the reviewer for the comments. In the revised manuscript, we have demonstrated that the drawn fibers can be treated as *textile fibers* and can be woven using the same equipment used to process conventional textile fibers. MEMS behavior is confirmed after the weaving process. To date, textiles serve primarily passive functions. Here, for the first time we demonstrate a MEMS textile system, in which the MEMS functionality is embedded directly at the fiber level. This is an enabling capability that could have many interesting applications in advanced functional textiles, including holographic display technologies^{1,2}, on-demand drug release³ for health and security purposes, etc. We discussed this potential of the fiber and fabric MEMS concept in different scenarios in the revised manuscript.

¹ Takaki, Y., Holographic 3D display using MEMS spatial light modulator (2012).

² Pearson, E.L., Mems spatial light modulator for holographic displays (2001).

³ Liu, C., Recent developments in polymer MEMS (2007).

“We note several potential future research directions and potential applications based on the unique MEMS fiber features. First, the axially-symmetric and indefinitely long fiber platform can open the door for MEMS textiles, initiating novel opportunities in advanced functional fabrics, such as holographic display technologies. Furthermore, electrically controllable microfluidic pumps can be enabled by integrating electrostrictive polymer devices with microfluidic channels in fibers, which present novel on-demand material (e.g. drug, solvent) release schemes in textiles. Secondly, the realization of long, thin, and flexible fibers with electromechanical transduction capabilities could enable new sensing and actuation applications in inaccessible regions or over extended lengths. For instance, the demonstrated actuation mechanism can provide the ability to navigate through the branches of narrow lumens of the body. Furthermore, other functions (such as sensing) can be integrated into the same fiber while keeping the diameter of the device as low as required paves the way for the development of novel types of steerable catheters. Thirdly, the ability to assemble fiber devices into grids or arrays makes them particularly well-suited for the coverage of large non-planar surfaces, an important feature for applications such as solar energy harvesting systems. For instance, flexible fiber MEMS with low power consumption can be embedded into flexible polymeric solar panels with integrated solar tracking capabilities. Finally, monolithic integration of electrodes into the fiber enables straightforward connection with external electrical circuits. This could enable electrically controllable artificial muscles which resemble human muscle, especially if elastomeric materials are utilized in the cladding of fiber MEMS”

Reviewer #3:

This paper is well written, and the results are interesting and convincing. It is a continuation of long standing work by this team, in which quite materials of very different material properties are co-drawn in fibre form. It extends this approach to relaxor ferroelectric materials, and investigates the use of this technology for micro mechanical devices.

While the results are interesting- and potentially may have multiple applications, they are not "new physics". Given the past history of the group (which has been working on co-drawing very different materials since 2000), it is debatable that it is sufficiently groundbreaking/surprising to be published in Nature. It might be more appropriate to a more specialist journal (perhaps Nature Materials).

Response:

We thank the reviewer for the comments. Please note that we are submitting this manuscript to *Nature Communications* as we believe the novelty of the results will interest a broad readership.

A technical question: the authors note: "the electrostrictive strain in fiber form exhibits remarkably better performance under significantly lower electric fields" than has been observed previously. They attribute this to processing conditions during the draw, which is a reasonable assumption. This being so, and given the relatively low temperatures used in the draw, it would be interesting to comment on the stability of the effects over time/elevated temperatures/other conditions that may arise in an application.

Response:

This is a very interesting question. For a practical application, all the reported properties of the fiber are expected to be robust up to the lowest softening temperature among the materials making up the fiber. In our fiber, lowest glass transition temperature is the conductive polyethylene at 110°C. Concerning temporal stability, a major advantage of electrostriction over ferroelectric mechanisms is that it enables a reproducible, non-hysteretic response, and is robust against aging¹.

¹ Cross, L.E., et. al, Large electrostrictive effects in relaxor ferroelectrics. (1980).

We discuss this point more clearly in the manuscript.

REVIEWERS' COMMENTS:

Reviewer #1 (Remarks to the Author):

I am happy that the authors have addressed my comments and that the manuscript is suitable for publication in Nature Communications.

Reviewer #2 (Remarks to the Author):

This is generally a much better manuscript than the first version. My various points (as reviewer 2) have been at least partially answered, although I would like to see more work on the 'practicality' aspects in any future publication.

Reviewer #3 (Remarks to the Author):

This resubmitted manuscript is much more compelling, and I support it being published in Nature Communications.

*Reviewers' comments are in **bold**, the authors' responses are Roman.*

Reviewer #1 (Remarks to the Author):

I am happy that the authors have addressed my comments and that the manuscript is suitable for publication in Nature Communications.

Response:

We are happy that we have addressed all comments of this referee.

Reviewer #2 (Remarks to the Author):

This is generally a much better manuscript than the first version. My various points (as reviewer 2) have been at least partially answered, although I would like to see more work on the 'practicality' aspects in any future publication.

Response:

We are happy that we have addressed the comments of this referee. Future work can indeed focus on extending the applications of this novel fiber MEMS technology.

Reviewer #3 (Remarks to the Author):

This resubmitted manuscript is much more compelling, and I support it being published in Nature Communications.

Response:

We are happy that we have addressed all comments of this referee.